# Sonographic Assessment of Complex Ultrasound Morphology Adnexal Tumors in Pregnant Women with the Use of IOTA Simple Rules Risk and ADNEX Scoring Systems

**DOI:** 10.3390/diagnostics11030414

**Published:** 2021-02-28

**Authors:** Artur Czekierdowski, Norbert Stachowicz, Agata Smoleń, Tomasz Kluz, Tomasz Łoziński, Andrzej Miturski, Janusz Kraczkowski

**Affiliations:** 1Department of Gynecological Oncology and Gynecology, Medical University of Lublin, 20-081 Lublin, Poland; 2Chair and Department of Epidemiology and Clinical Research Methodology, Medical University of Lublin, 20-080 Lublin, Poland; norberts@umlub.pl (N.S.); absmolen@wp.pl (A.S.); 3Department of Obstetrics and Gynecology, Fryderyk Chopin University Hospital No 1, Faculty of Medicine, Rzeszow University, 35-060 Rzeszow, Poland; jtkluz@interia.pl; 4Department of Obstetrics and Gynecology, Pro-Familia Hospital, 35-302 Rzeszow, Poland; tomasz.lozinski@pro-familia.pl; 5Department of Obstetrics and Pathology of Pregnancy, Medical University of Lublin, 20-081 Lublin, Poland; andrzej.miturski@umlub.pl (A.M.); jacek.kraczkowski@umlub.pl (J.K.)

**Keywords:** ovarian, tumor, ultrasound, pregnancy, simple rules risk, ADNEX, ROMA algorithm

## Abstract

Background: To evaluate the accuracy of subjective assessment (SA), the International Ovarian Tumor Analysis (IOTA) group Simple Rules Risk (SRR) and the Assessment of Different NEoplasias in the adneXa (ADNEX) model for the preoperative differentiation of adnexal masses in pregnant women. Methods: The study population comprised 36 pregnant women (median age: 28.5 years old, range: 20–42 years old) with a mean gestation age of 13.5 (range: 8–31) weeks at diagnosis. Tumors were prospectively classified by local sonographers as probably benign or probably malignant using SA. Final tumor histological diagnosis was used as the reference standard in all cases. Logistic regression SRR and ADNEX models were used to obtain a risk score for every case. Serum CA125 and human epidydimis protein 4 (HE4) concentrations were also retrieved and the Risk of Ovarian Malignancy Algorithm (ROMA) value was calculated. The calculated predictive values included positive and negative likelihood ratios of ultrasound and biochemical tests. Results: Final histology confirmed 27 benign and 9 malignant (including 2 borderline) masses. The highest sensitivity (89%) and specificity (70%) were found for the subjective tumor assessment. Although no malignancy was classified as benign using the SRR criteria (sensitivity = 100%), the specificity of this scoring system was only 37%. At the cut-off risk level of >20%, the ADNEX model had a sensitivity of 78% and a specificity of 70%. Serum levels of CA125, HE4 and the ROMA risk model correctly identified adnexal malignant tumors with a sensitivity of 67%, 25% and 25%, respectively. Corresponding specificities were 72%, 100% and 100%, respectively. The highest positive and negative likelihood ratios were found for SA (LR+ = 3.0 and LR− = 0.16, respectively). Overall diagnostic accuracy of all predictive methods used in this study were similar (range: 70–75%) except for SRR (53%). Conclusion: Subjective assessment remains the best predictive method in complex adnexal masses found at prenatal ultrasound in pregnant women. For less experienced sonographers, both the SRR and ADNEX scoring systems may be also used for the characterization of such tumors, while serum tumor markers CA125 and HE4, along with the ROMA algorithm appear to be less accurate.

## 1. Introduction

The extensive use of ultrasound in early pregnancy has led to an increased detection of ovarian lesions that are diagnosed incidentally in asymptomatic gravid women [1,2]. When pelvic mass is suspected in a pregnant woman, transvaginal sonography is currently the first imaging modality for the adnexal lesion discrimination. In cases of larger masses, transabdominal sonography is also used. The current diagnostic challenge in pregnant women with isolated adnexal tumors is related to the appropriate differentiation of these masses in order to offer the best available treatment pathways [3]. The most common pregnancy-related adnexal masses are simple cysts, the corpus luteum cysts and the theca lutein cysts. All of these lesions may resolve late in the second trimester of gestation, although in some women they can persist until after delivery. Most adnexal masses detected in early pregnancy are benign, and sonographers do not need complicated scoring systems or predictive models to classify the type of tumor correctly. Ovarian cancer is the second most common gynecological cancer reported to coexist with pregnancy, and it is estimated that 1 in 15,000 to 1 in 32,000 women are affected [4]. However, the prevalence of malignancy among ovarian masses diagnosed in pregnant women varies from 0 to 9% [5]. When possible, surgical excision of complex masses is indicated, typically at the beginning of the second trimester of gestation. Minimally invasive and fertility-sparing surgery can be offered to suspected cases of borderline tumors, stage I epithelial ovarian cancers (EOC) according to the International Federation of Gynecologists and Obstetricians (FIGO) classification, as well as selected cases of germ cell or sex-cord stromal ovarian tumors [6].

The diagnostic use of serum tumor markers in women with adnexal masses found in pregnancy remains controversial. Pregnant women in the first trimester showed higher CA 125 serum levels that the non-pregnant controls. In contrast, the concentrations of CA 125 in the second and third trimester and in the control group were comparable [7]. Although CA125 and another popular tumor marker, Human Epidydimis protein 4 (HE4), were assayed in pregnant women, the prognostic value of those markers used alone or in combination (ROMA test, i.e., the Risk of Malignancy Algorithm) in discriminating adnexal tumors in pregnancy is unknown [8,9,10].

At ultrasound examination lesion size and morphology are the key sonographic features used to triage women with adnexal masses for expectant or interventional management. Outside pregnancy pattern recognition based on expert subjective assessment of adnexal tumor features remains the best method that can be used to preoperatively distinguish the type of ovarian mass [11]. For sonographers who are not expert ultrasound users, other prognostic methods are available and include the GI-RADS system, International Ovarian Tumor Analysis (IOTA) group proposed Simple Descriptors (SD), Logistic Regression models 1 and 2 (LR1 and LR2) and the Simple Rules (SR) [12,13]. The Simple Descriptors method cannot practically be used in pregnant women as one of the requirements for a malignant descriptor is the patient’s age over 50 years [11]. The SR utilizes the presence or absence of the selected and specific tumor features such as size, locularity and solid parts’ presence, color Doppler blood flow and evidence of free fluid outside the pelvis [14]. In non-pregnant patients, this method when used for sonographic characterization of ovarian tumors had 78% specificity and 87% sensitivity [15]. However, there are important limitations to the use of the original SR scoring system. An adnexal lesion is characterized as benign when only B (benign) features are observed, and the lesion is classified as malignant when only M (malignant) features are present at ultrasound exam. When the adnexal tumor has no B and M features or both types of features are present, the mass should be considered as indeterminate [16]. However, other scoring systems developed by the IOTA group over recent years may be also used to stratify the risk of malignancy in virtually all adnexal masses. These methods include the Assessment of Different NEoplasias in the adneXa (ADNEX) model [17] introduced in 2014 and the Simple Rules Risk (SRR) which was proposed in 2016 [18]. Despite multiple validation studies that confirmed their high predictive value, to the best of our knowledge neither of the models was used so far for the discrimination of complex adnexal tumors detected in pregnant women at ultrasound examination [19,20,21]. The aim of this study was to compare the subjective tumor assessment (SA) with the use of SRR and ADNEX scoring systems in the presurgical discrimination of complex adnexal masses in pregnancy. Additionally, we have assessed the predictive value of commonly used ovarian tumor markers CA125 and HE4 along with the ROMA algorithm for the discrimination of adnexal masses in pregnancy.

## 2. Material and Methods

### 2.1. Study Design and Participants

This was a retrospective multicenter study carried out in two departments of obstetrics and gynecology (Departments of Obstetrics and Gynecology of the Fryderyk Chopin University Hospital No 1 and of the ProFamilia Hospital, Rzeszow, Poland), one Department of Obstetrics and Pathological Pregnancy (Medical University of Lublin, Poland) and one Gynecological Oncology Center (Medical University of Lublin, Poland). From the clinical records of participating centers, we identified all pregnant women with an ultrasound diagnosis of an adnexal mass detected before delivery between September 2013 and December 2020. At the time of ultrasound examination, a standardized protocol was filled out in which the tumors were classified as probably benign or malignant using subjective assessment of referring sonographer. The inclusion criteria were (1) a complex adnexal mass with a good quality ultrasound pictures/video clips and (2) histological diagnosis of the tumor type made during pregnancy or at cesarean section. For the purpose of this study, patients with simple adnexal cysts and ovarian corpus luteum cysts were excluded because these masses are typically easy to identify at ultrasound examination and surgery in such cases during pregnancy is extremely rarely indicated. Tumors were also excluded if the ultrasound information was not sufficient. Next, all imaging reports of adnexal masses with complex morphology were analyzed retrospectively with the knowledge of final tumor histological diagnosis. In particular, we searched for references to findings that were, or could have been, related to the increased risk malignancy. Sonographic tumor features were analyzed in relation to whether the cases were diagnosed correctly. Clinical, sonographic and histological parameters, as well as information on the type of surgery (laparoscopy vs open surgery) and gestational week when surgery was performed were retrospectively retrieved from the patients’ medical records. Indications for surgery according to the referring clinicians’ decision were recorded in terms of suspicion of malignancy based on specialist’s opinion. In order to be able to classify all lesions based on the Simple Rules assessment, we applied retrospectively the IOTA logistic regression model Simple Rules Risk (SRR) and IOTA ADNEX model to obtain a risk score for every case. We used a cut-off of 20% for both the SRR and ADNEX-calculated probability to define women with suspected malignancy. Indications for surgery included suspicion of malignancy or high risk of side effects such as torsion, rupture or obstacle to normal full-term pregnancy. Patients with adnexal lesions considered to be benign but presenting with symptoms thought to be attributable to the lesion were referred to surgical management by their obstetricians. Pregnant women with an adnexal mass suspected to be malignant were referred to the gynecological oncology unit. Clinical, ultrasound, surgical and histological parameters, as well as information on the type of management were retrospectively retrieved from the databases of the ultrasound units and from patients’ medical records. The reference standard was histological diagnosis of the tumor type. For statistical purposes, borderline tumors were categorized into the malignant group.

### 2.2. Ultrasound Assessment

All patients were examined with transvaginal and, if necessary, also with transabdominal ultrasound. The patients were scanned in ultrasound units of the four participating centers by sonographers who were gynecologists with an experience equivalent to Level 2 of training according to the European Federation of Societies for Ultrasound in Medicine and Biology (EFSUMB) for gynecological ultrasound [22]. The expert sonographer who performed subjective assessment (SA) was a gynecologist with training and experience equivalent to Level 3 of training according to the EFSUMB for gynecological ultrasound, which means “a work in a reference tertiary center, an academic record, a high level of experience and expertise and the majority of the time undertaking gynecological ultrasound and/or teaching, research and development” [22]. 

Ultrasound imaging was carried out using high-end ultrasound equipment: Voluson S10, E8 or E10 (GE Healthcare, Zipf, Austria) or Samsung WS80A or Hera W10 (Samsung, Seoul, Korea). The frequency of the vaginal probes varied between 5.0 and 9.0 MHz and that of the abdominal probes between 3.5 and 5.0 MHz. All masses were described using the International Ovarian Tumor Analysis (IOTA) terminology [23]. The following parameters were assessed: location of the lesion, size of the lesion (three orthogonal diameters), unilateral or bilateral mass, presence of ascites and/or fluid in the pouch of Douglas, type of mass (unilocular, unilocular–solid, multilocular, multilocular–solid or solid), presence of papillary projections (defined as any solid protrusion into a cyst cavity with a height of ≥3 mm), number of papillary projections within the cyst, irregularity of the surface of papillary projections, presence of solid tissue other than papillary projections and presence of septa. In the case of bilateral masses, the mass with the most complex ultrasound morphology was used. If the masses had similar morphology, the larger mass was used. The color content of the papillary projections and of the solid tissue other than papillary projections at power Doppler examination was subjectively estimated, using a color score (1 = no vascularization; 2 = minimal vascularization; 3 = moderate vascularization; and 4 = strong vascularization) [23].

Subjective assessment by an expert sonographer classified ovarian masses into three categories: benign, borderline and malignant. Level 2 sonographers used the SRR and ADNEX model to classify all adnexal masses. Details of SRR and ADNEX models are presented in the Appendix A. For the purpose of this study, we used the SRR calculator that is freely available on the IOTA group website [24].

For statistical purposes, we used 3 categories of SRR risk: low (<3%) intermediate (3–20%) and high (>20%). The IOTA-ADNEX is the first polytomous predictive system for adnexal masses [18]. ADNEX uses the input of three clinical variables, including age (years), serum CA-125 level (U/mL, the use of this marker serum concentration is optional) and type of center (oncology center vs. non-oncology center), and the measurement of six sonographic variables, including the maximal diameter of the lesion (mm), maximum diameter of the solid tissue, number of papillary projections (5 categories: 0/1/2/3/>3), more than 10 cyst locules, presence or absence of acoustic shadow and presence or absence of extra pelvic ascites. In the type of center, an “oncology center” is defined as a tertiary clinical center with specific gynecologic oncology, and the rate of ovarian malignancy in oncology centers and other hospitals is usually “between 22–66%” and “below 30%” [18].

By inserting relevant clinical and sonographic features into the formula, the probability of benign and malignant tumor was calculated. The patient’s risk was expressed as a relative risk (RR) in comparison with the baseline risk with the prescribed formula. The model presented five categories: benign, borderline tumors, FIGO stage I invasive, FIGO stage II–IV invasive ovarian cancer and secondary metastatic cancer. The resulting graph showed the risk percentage for each category, and markers for large values of relative risks (RR) are highlighted. Following calculation of RR with the ADNEX model, the adnexal lesions were grouped into low (<3% risk), intermediate (3–20% risk) and high-risk (>20%) categories. According to the results of the IOTA group’s recent studies, a cut-off risk set at 20% is related to the best balance between ADNEX model sensitivity and specificity [17]. All principles related to the ADNEX model were followed by the practical guidelines [25]. We used the ADNEX risk calculator that is available on the IOTA group website [26]

### 2.3. Tumor Markers

The measurements of serum CA125 and HE4 levels were determined on Cobas e601 analyzer from Roche Diagnostics (Basel, Switzerland) using an electrochemiluminescence immunoassay (ECLIA). For CA125, a standard recommended cut-off value of 35 U/mL was used and for HE4, since all women were premenopausal, a cut off value of 70 pmol/L was used. In women who had both CA125 and HE4, serum preoperative concentrations available the risk of malignancy were calculated using the online ROMA risk calculator [27]. The probability of the ovarian malignancy cut-off value calculated with the ROMA test was set at >11.4%, which is typically recommended for premenopausal women [28]. ROMA algorithm calculation formula is presented in the Appendix A.

### 2.4. Statistical Analysis

All data were entered into a Microsoft Excel 2016 spreadsheet, and the dedicated database was constructed for the purpose of this study (Appendix A). Data were expressed as median and range or n (percentage) according to their type. Continuous data were shown as median and range. The statistical significance of differences between categorical data was determined using the Chi-squared test, while differences in continuous and ranked data were determined using the Mann–Whitney test. *p* values of < 0.05 were considered to be statistically significant. Diagnostic performance measures of all three strategies (SA, SRR and ADNEX model) were computed. Reported performance measures were sensitivity and specificity, assuming that surgical intervention was indicated in intermediate or high-risk lesions. The positive (LR+) and negative (LR−) likelihood ratios and overall accuracy of the three strategies were also calculated. In all calculations, the software package Statistica v.10.0 (Statsoft Tulsa, OK, USA) was used.

## 3. Results

From the databases of 4 participating centers we identified 36 pregnant women that fulfilled our study inclusion criteria. In 35 cases, adnexal tumors were removed via laparoscopy or laparotomy, and in one case the diagnosis of the metastatic tumor was made with the tumor biopsy. Adnexal tumor was removed during cesarean section in 20 patients. Histological diagnosis of all masses is presented in Table 1. 

Of the 36 study participants, 27 (75%) had benign tumors and 9 (25%) had malignant/borderline adnexal lesions. The most commonly found benign tumors were dermoid cysts (11 cases) and decidualized endometriomas (10 cases). Of the nine malignant masses, three women had serous or endometrioid invasive ovarian cancer and two patients had borderline ovarian tumors (BOTs). There was also one case of ovarian dysgerminoma and two cases of ovarian metastases from colorectal cancer.

Selected demographic and clinical data are shown in Table 2. 

Median age of pregnant women with benign and malignant tumors was 28 (range: 20–42) years old and 29 (range: 24–41) years old, respectively. There were 22 (61%) nulliparous women, and the proportions of nulliparas were comparable between both groups. The median gestational age at tumor sonographic detection was 13.5 (range: 8–31) weeks and the median gestational age when surgical procedure was performed was 21.5 (range: 12–40) weeks. At the time of data collection, four pregnancies were ongoing. In the remaining group, vaginal term delivery occurred in 8 of 27 (29.6%) women with benign masses and in 4 of 8 (50%) patients with malignant tumors. All except one operated woman during the course of their pregnancies had no postsurgical fetal or maternal complications. The only case of fetal exitus and maternal death one month after surgery at 28 weeks was related to stage IV metastatic colorectal cancer. 

Sonographic characteristics of the studied tumors are shown in Table 3. The median adnexal lesion diameters were 71 mm (range 39–206 mm) and 90 mm (range 45–135 mm) in the benign and in the malignant tumors, respectively. There were eight unilocular cysts with complex morphology found at ultrasound examination and all were benign. A total of 6 out of 9 malignant masses and 15 out of 27 benign masses had either unilocular–solid or multilocular–solid cyst appearance. Both borderline tumors were described as unilocular-solid masses. If present, the median size of the largest solid component found in malignant masses was more than twice higher than that found in benign tumors (39 mm vs 18 mm). Of five masses classified as solid, two were benign and three were malignant. Benign and malignant masses differed significantly in the maximum size of the solid component (*p* = 0.01). Eleven (40.7%) benign tumors and five out of nine (55%) malignant masses had one or more internal cyst wall papillary projection(s). The median height of largest tumor papillary projection was 14.5 mm (range np. 3–42 mm). In all but one case of benign tumors, these papillae were smooth shaped, whereas the smooth contour was also detected in three out of five malignant masses. In one case of high grade endometrioid cancer, the papillations were smooth shaped and vascularized. In the remaining three invasive tumors, their papillations had uneven contour and were vascularized in all cases. 

Selected morphological characteristics of the studied adnexal masses are presented in Table 3. 

Figure 1 shows transvaginal sonographic and intraoperative images of a 90 mm maximum diameter cystic–solid mass with mixed fluid echogenicity and multiple papillary projections.

The mass was wrongly classified as a benign dermoid cyst at ultrasound examinations performed in different ultrasound centers at 8 and 12 weeks of gestation. The patient was correctly diagnosed by the pattern recognition method at 23 weeks of gestation; however, a midline vertical laparotomy had to be postponed until 26 weeks due to existing comorbidities. Although both tumor marker serum concentrations were only slightly elevated (CA125 = 35.43 IU/mL, HE4 = 75.9 pmol/L), the ROMA test performed before surgery indicated a high risk of ovarian malignancy (20.1%). Midline vertical laparotomy was performed and revealed a 9-cm-large tumor with a small metastatic solid mass between the enlarged gravid uterus and the main tumor capsule. This metastatic lesion beyond the main tumor could not be detected at prenatal ultrasound late in the second trimester. Serous ovarian cancer, FIGO stage IIC, was confirmed on final histological examination. The postoperative course was uncomplicated, and at the time of writing the pregnancy is ongoing and the patient is scheduled for chemotherapy.

At color Doppler assessment, all malignant tumors were vascularized (IOTA color score 2–4), but also 13 (48.2%) of benign masses exhibited similar type of vascularization. The difficulties with the correct vascularity assessment in benign and malignant tumors are illustrated in a case of a 37-year-old pregnant woman diagnosed at 17 weeks of gestation with a solid cystic mass (Figure 2). The lesion was in some parts strongly vascularized, and multiple irregularly shaped papillary projections were seen. Subjective assessment indicated that the mass was highly suspicious for ovarian invasive cancer. Serum CA125 was elevated (63.7 U/mL), and serum HE4 was normal (43 pmol/L). The calculated ROMA risk was low (7.3%). The ADNEX model indicated a very high risk of malignancy of 89.5%, and the use of SRR method resulted in an 89.2% probability of malignancy. However, at 22 weeks of gestation following laparoscopic removal of the mass with the use of a glove endobag, final histology confirmed a benign decidualized endometrioma.

Figure 3 presents a small (38 mm in maximum diameter) solid ovarian lesion that was detected at eight weeks of gestation, and due to fast tumor growth, was operated on at 15 weeks of gestation. Preoperative serum concentrations of CA125 = 97 U/mL and HE4 = 39.8 pmol/L were recorded. The ROMA test result indicated a low risk of malignancy of 4.5%. However, the presence of two “M” features (M3 and M5) assessed in the SRR method clearly indicated a high risk of malignancy that was calculated to be 94.1%. The ADNEX model risk was only 4.4%. Serous ovarian cancer FIGO stage IC was confirmed on final histology.

Additional other tumor features such as the presence of blood flow in papillae, the presence of shadows beyond papillation or tumor wall, the presence of microcystic pattern of solid parts and vascularization of solid parts were also recorded. All these features did not differ significantly between patients with benign and malignant tumors. Extraovarian metastases were seen in both cases of metastatic tumors, and ascites was found in only one case of malignant tumor. 

Serum CA125 concentrations were available in 75% of women, and serum HE4 levels were available in 66.7% of cases. The ROMA risk could be calculated only in this group of our study participants. 

Table 4 presents the distribution of the prognostic tests results in benign and malignant tumors.

Significant differences between studied groups of women with benign and malignant adnexal masses were found for nearly all tests used. When the ADNEX risk was elevated by >20%, the highest risk was found most frequently for borderline ovarian tumors. Interestingly, in both studied pregnant women with borderline ovarian tumors (BOTs), serum concentrations of tumor markers were not elevated (CA125 = 27 U/mL and 22.45 U/mL, and HE4 = 47.6 and 49.6 pmol/L, respectively). The corresponding ROMA risks for both patients were accordingly below the 11.4% risk threshold. Additionally, in one of our patients with solid vascularized ovarian tumor that was confirmed to be colorectal metastatic cancer, both serum CA125 (24.4 U/mL) and (HE4 (29.9 pmol/L) levels were within normal limits.

Table 5 presents the numbers of cases correctly or incorrectly classified in both groups by various tests with the calculated likelihood ratios and prognostic accuracy of these tests.

Using the SRR criteria, no malignancy was classified as benign (sensitivity = 100%) in our studied group; the specificity of this scoring system was only 37%. At the cut-off risk level of >20%, the ADNEX model had a sensitivity of 78% and a specificity of 70%. Risk models of serum levels of CA125, HE4 and ROMA correctly identified adnexal malignant tumors with a sensitivity of 67%, 25% and 25%, respectively. Corresponding specificities for these tests were as follows: 72%, 100% and 100%, respectively. The highest positive and negative likelihood ratios were found for SA (LR+ = 3.0 and LR− = 0.16, respectively). Overall diagnostic accuracy of all predictive methods used in this study were similar (range: 70–75%) except for the SRR (53%). Figure 4 presents an ovarian lesion that was difficult to classify by both expert subjective assessment and the SRR model. 

## 4. Discussion

We have described ultrasound features, management and outcomes in 36 pregnant patients with complex adnexal masses detected at ultrasound examination. We have found that the most difficult for correct classification lesions were unilocular or multilocular cysts presenting with papillary projections and/or solid parts. Ovarian tumors may be found at early pregnancy ultrasound scans performed for non-gynecological purposes. Persistent and, especially, adnexal lesions with complex morphology found at ultrasound examination, even when asymptomatic, must be carefully assessed to exclude possible ovarian cancer. Although rare, other malignancies such as dysgerminoma, granulosa cell tumor or metastatic lesions originating mainly in the GI tract must be ruled out in the clinical setting of pregnancy, especially in younger women.

Our results confirm that the differentiation between benign and malignant masses found in pregnant patients may be difficult for sonographers with different background training, e.g., for obstetricians and radiologists. Our study also demonstrates some diagnostic pitfalls reflected by the occurrence of adnexal tumor non-typical morphology. The same types of difficulties were reported by Meys et al. [29]. The sonographic appearance of some ovarian lesions may vary according to pregnancy status. In our study, most different histological adnexal entities showed predictable sonographic features; however, our data also demonstrate that for each histological entity there are some cases that do not exhibit typical features; for example, absence of an irregular contour of papillae in invasive ovarian cancer or lack of a microcystic pattern in borderline tumors. In pregnant women the appearance of adnexal mass with non-typical morphology may be due to physical limitations. For example, there might be an absence of Doppler signals when the distance between the ultrasound probe and the lesion is large, or some important tumor features may not be visible when gravid uterus is obscuring the field of view. In our study, when papillary intracystic projections were present in benign masses, they were all smooth shaped except in one case of cystadenofibroma. Only a minority of decidualized endometriotic cysts exhibited regular shaped papillary projections, most being irregularly shape at ultrasound examination. Moreover, a smooth papillary projection contour was also detected in three out of five malignant masses. In our series, we had one such case when invasive ovarian cancer presenting as a cystic–solid mass was misclassified by a Level 2 sonographer at 12 weeks of gestation and on several more occasions as decidualized endometrioma (Figure 4).

Several various features of papillary projections, including, e.g., number, maximum size, external contour shape, the so called “microcystic pattern”, vascularity and shadows behind the papillae, have been recently proposed and verified in clinical studies performed outside gestation [30,31]. The growth pattern of papillary projections can be endophytic, exophytic or mixed. In the vast majority of cases, ultrasound imaging is unable to show small exophytic papillary growth unless ascites is present. Our case of a small, early stage solid cancer presented in Figure 3 illustrates these difficulties. 

Recently, Li et al. [32] presented the results of their MRI study on papillary projections patterns in BOTs and invasive ovarian cancers. They have found that of the three types: endophytic, exophytic and mixed, more than half of the papillae or nodules in BOTs and invasive masses were endophytic. Among those with exophytic or mixed pattern, the exophytic type was observed predominantly in BOTs, while the mixed pattern was found mainly in invasive cancers. Moro et al. described recently most characteristic features of malignant ovarian tumors in 22 pregnant patients diagnosed with ultrasound and treated in a single institution over 20 years [33]. In accordance with our results presented above, their studied serous borderline tumors were unilocular–solid or multilocular–solid lesions, whereas mucinous borderline tumors were multilocular or multilocular–solid masses. All serous/endocervical-type borderline tumors had papillary projections. Most invasive epithelial ovarian cancers were multilocular–solid masses. In Moro et al. study, all metastatic tumors were solid masses at ultrasound examination. This was in contrast to our findings where two solid masses (one dysgerminoma and one small serous cancer) were primary ovarian malignancies, whereas our two metastatic tumors were characterized as cystic–solid lesions. Both in that study and in our study population, no maternal or neonatal complications were reported for patients operated because of borderline tumors or epithelial ovarian carcinomas. Moro et al. have concluded that morphological features of malignant ovarian masses detected during pregnancy at ultrasound examination were similar to those described in non-pregnant patients [33]. Unfortunately, that study has not used any scoring systems to asses ovarian tumor type, nor did it present sonographic data on other potentially difficult-to-discriminate benign ovarian masses found in gestation.

Webb et al. [34] reviewed reports on 563 adnexal masses in pregnancy and found that the incidence of these lesions varied from 0.1 to 2.4%, with an average of 0.02%. Regarding the likelihood of malignancy, 48% were classified as simple and 52% as complex. Among the simple masses 1% was malignant, while among the complex masses 9% were malignant. Testa et al. [35] have recently presented an analysis of 113 pregnant women with adnexal tumors of whom 56 were operated during the course of gestation. Using the IOTA group ultrasound morphological features, no malignancy was found in 37 women with unilocular cysts. In contrast, the prevalence of malignant masses was 27% in the multilocular group, 35% in the unilocular–solid group, 70% in the multilocular–solid group and 70% in the solid group. Neither obstetric nor neonatal complications were reported for patients in the surveillance group or in those with benign, borderline or primary epithelial invasive histology. These results are in line with our findings. Both in our study and in Testa et al.’s study, maternal deaths (one and three, respectively) were observed in patients with ovarian metastases.

In the absence of expert opinion, various mathematical predictive models used for adnexal tumor discrimination demonstrate an advantage over subjective assessment due to their relative simplicity and potential usefulness for less experienced sonographers. However, it is currently unclear which predictive model would be the best at predicting the nature of an ovarian tumor in pregnant women. In our study, the SR scoring system was proved to be almost as good as subjective expert examination; however, the limitations of this method in adnexal tumors diagnosed during pregnancy are not known. 

Despite initial optimistic multicenter verification studies showing that only minimal expertise was required to use the SR scoring system [36], the study of Knafel et al. presented conflicting results [37]. These authors studied the use of IOTA SR in non-pregnant patients and found that the number of inconclusive cases was likely to depend on the level of ultrasound expertise. Moreover, less experienced examiners tended to overestimate blood flow and a presence of acoustic shadows within the tumors. The majority of inconclusive cases comprised malignant masses that were either unilocular–solid, solid or multilocular–solid lesions with a maximum diameter of less than 100 mm. We observed the same difficulties with correct tumor classification by Level 2 sonographers. One of our cases when a malignant tumor was correctly suspected with both SA by an expert examiner and SRR was a solid mass of a 44 mm maximum diameter. Laparoscopic surgery performed at 16 weeks of gestation revealed low grade serous ovarian cancer FIGO stage IC. Apart from the sonographer’s experience there are other important issues related to the potential SR method use in pregnant patients. Both malignant and benign cystic–solid lesions may have more than four papillary projections. Apparently, one of the malignant Simple Rules criterion (M3) is present in such cases, and the mass should be classified as malignant (the SRR risk is 63.6%). The same situation happens when abundant vascularization in tumor papillae or other solid part is detected at color Doppler examination. Again, another M criterion is present (M5), and according to the original SR method and in the absence of other B and M features, the mass should be classified as malignant. Moreover, the calculated risk of malignancy with the SRR method is 64%. When both M3 and M5 features are present in one tumor, then calculated SRR is 89.2% [18]. 

Complications of ovarian endometriosis during pregnancy are rare, and there is no evidence that the disease could have a major negative influence on pregnancy outcome. However, pregnancy associated hormonal effects on endometriotic cysts may initiate the process of ectopic decidualization and the formation of papillary projections within the cyst wall [38]. Some of these papillations may be strongly vascularized at ultrasound color Doppler examination. According to several recent publications, the described adnexal masses are in most cases easy to distinguish from borderline or invasive ovarian cancers [39]. Mascillini et al. [40] studied 34 cases of ovarian cysts with papillary projections detected in pregnant women. The have found that ovarian cysts with ground-glass echogenicity and papillations with a smooth contour on ultrasound are most likely to be decidualized endometriomas. On the contrary, in cysts with anechoic or low-level echogenicity and papillations with an irregular contour, borderline histology should be considered as the most likely. Our observations indicate that the appearance of papillary projections and other solid parts may change during the course of gestation, and sonographic findings must be interpreted with caution. Awareness of this process can help prevent misdiagnosis of decidualized endometriomas as ovarian malignancy and false negative diagnosis of endometrioma when, in fact, malignant ovarian tumor is present. 

In our series, the most difficult for a correct classification masses were decidualized endometriomas with 6 out of 10 cases suspected to be malignant on both subjective assessment and the SRR scoring system. Additionally, one case of ovarian endometriod cancer presented in Figure 4 had only smooth-shaped non-vascularized papillary projections. The lesion was wrongly classified by a Level 2 sonographer as benign and observed through the course of gestation. If the SR criteria had been used, the presence of one M feature (M3) should suggest a high risk of malignancy. The mass was removed at cesarean section that was performed at 37 weeks of gestation for fetal indications. Unfortunately, the capsule of this tumor was damaged during fetal delivery. Tumor capsule rapture was the most likely reason of a subsequent fast intra-abdominal cancer spread, and apart from chemotherapy the patient required a second debulking surgical procedure a few months later. Frühauf et al. have recently suggested [41] that if ultrasound Simple Rules are not applicable or show probable malignancy, the pregnant patient should be referred to a tertiary center for expert ultrasound assessment. For such difficult cases, they would recommend magnetic resonance diffusion weighted imaging (MRI-DWI) suggesting that this method would be more accurate. MRI-DWI may distinguish with high accuracy products of blood degradation and may recognize lower tissue cellularity of benign decidualized endometriomas in comparison to invasive malignant ovarian tumors [42]. However, MRI is not widely available for pregnant women, and pelvic examination requires a high level of expertise in the diagnosis of gynecological malignant masses.

To our knowledge, this study represents the first application of the IOTA SRR scoring system and the ADNEX model in pregnant women. In our patients, the use of SRR allowed the correct preoperative classification of all nine malignant tumors. However, the sensitivity of this test was relatively low (37%). To date, to the best of our knowledge only one study that estimated the role of SRR in non-pregnant women with adnexal masses was published. Hidalgo et al. [43] used the three-step and the two-step strategy to assess adnexal masses with pelvic sonography. In the two-step strategy using simple descriptors (SD) as a first step and SRR in the second step, the sensitivity and specificity of this two-step strategy were 98.4% and 63.8%, respectively. Similar to SRR, the ADNEX model has not yet been used in the discrimination of adnexal masses found in gestation. It is surprising given the fact that the high predictive values of this scoring system have been confirmed in multiple studies in non-pregnant women [21,44,45]. In contrast, in our specific group of complex adnexal masses, the ADNEX model had a sensitivity of 78% and a specificity of 70%. When malignancy in ovarian or adnexal mass is suspected in pregnant woman, a high sensitivity of the test is required because of serious consequences of the wrong or delayed diagnosis. However, a high specificity is also of utmost importance as many adnexal lesions are detected incidentally at prenatal ultrasound scanning in asymptomatic patients. A dominant role of ultrasonography in adnexal masses evaluation both in pregnant and non-pregnant women is related to its specificity, allowing confident diagnosis in the majority of benign adnexal lesions.

As in non-pregnant women with adnexal tumors, the main problem with using biomarkers or biomarker algorithms such as the ROMA test as a diagnostic tool to predict malignancy in isolated ovarian masses is their rather poor sensitivity. In particular, the most commonly used ovarian cancer marker, CA125, has typically increased serum concentrations during the first trimester, whereas in the second and third trimesters its levels are low in the maternal serum but may be high in the amniotic fluid [7]. HE4 is a relatively novel tumor marker approved in 2011 by FDA for the detection of ovarian cancer and monitoring the recurrence or disease progression in conjunction with CA125 [8]. To date, there are only few studies that examined HE4 serum concentrations in pregnant women. Wang et al. have recently found that both HE4 and CA125 levels in the third trimester were significantly higher than in the control group [9]. There was no difference in HE4 between the control, the first and the second trimester groups, while levels of CA125 in first trimester group were significantly elevated. Uslu et al. examined longitudinally the physiological changes of serum HE4 concentrations at each trimester of pregnancy [10]. In that study, the median HE4 concentrations decreased during first and second trimesters of pregnancy and did not change during third trimester of pregnancy. Moreover, pregnancy and several other factors were reported to strongly influence CA125 concentrations; however, those factors produce no or minimal effect on HE4 serum levels that in turn are more age-dependent than CA125 [46]. Lu et al. have studied changes of CA125 and HE4 levels and presented their reference intervals for the ROMA index in the sera of pregnant women without adnexal masses [47]. Our results indicate that the diagnostic accuracy of serum levels of CA125, HE4 and the ROMA test in complex adnexal tumors found in pregnant women at ultrasound examination was much lower than that of ultrasound-based methods. Our results clearly show that borderline tumors, and less frequently early invasive cancers and even adnexal metastatic tumors, may present with normal CA125 and HE4 concentrations in blood serum and with low ROMA risk values. 

In non-pregnant women, Valentin et al. have found that adding a single CA125 measurement to ultrasound imaging performed by an experienced examiner does not improve preoperative discrimination between benign and malignant adnexal masses [48]. Dochez et al. were the first to use the strict IOTA group criteria for presumed benign ovarian tumors (PBOTs) and compared the predictive results with the ROMA algorithm [49]. The Authors have found that the combination of elevated HE4 and CA125 was the best tool to confirm the risk of ovarian cancer in patients with PBOTs. In that study, a combination of tumor markers used as the ROMA algorithm had a specificity of 99.5% and LR+ of 104.5, which was better than the estimation of each serum marker expression alone. However, the results of that multicenter study must be interpreted with caution as there were only 12 malignant lesions among the 221 included non-pregnant women which resulted in a rather low maligant tumor prevalence of 5.4%.

Although both IOTA predictive models were designed with the use of only non-pregnant women, it was tempting to check the potential usefulness of both methods in adnexal masses diagnosed during the course of gestation. As most ovarian lesions are probably examined by sonographers or specialists who are not experts in gynecologic ultrasonography (Level 2), it seems reasonable to suggest that our findings could offer some additional clues and ideas on the performance of the different adnexal tumor risk models in daily clinical practice. Our results indicate that both SRR and ADNEX may be used with caution in the management of patients with adnexal masses detected during pregnancy. Despite known limitations, subjective tumor assessment at ultrasound examination seems to be the best method of complex adnexal tumor differentiation in pregnant women. 

Regardless of the SRR or ADNEX model results or tumor marker levels, when a unilocular–solid cyst with papillary projections increases in size and in number of papillations from the first trimester during pregnancy, an invasive tumor cannot be excluded and surgical exploration should be considered. In general, in multilocular–solid or purely solid masses, the risk of malignancy is much higher and immediate surgical management, preferably at a gynecological oncology center, should be considered [50]. 

## 5. Strengths and Limitations

We described for the first time the predictive value of the SRR and ADNEX scoring systems, as well as CA125 and HE4 with the ROMA algorithm in pregnant patients with complex adnexal masses. Multiple sonographic tumor features were described using IOTA terminology. All masses were histologically verified and the proportion of malignant tumors (9/36, 25%) was relatively high. The main limitations of our study were the retrospective nature and the relatively low number of participants included. Because of that, meaningful comparisons between the groups of patients with benign and malignant masses were difficult to perform. The differences in sonographers’ experience and ultrasound systems settings could have contributed to the image quality at the time of their acquisition. Another limitation was that some information on important clinical and biochemical parameters was missing. Moreover, despite meticulous analysis of ultrasound images, the video clips from the ultrasound examination were not always available. Our ability to revise image assessment and eventually detect typical ultrasound features of adnexal tumors was in this way limited.

In summary, we conclude that adnexal masses in pregnant women that are complex at ultrasound examination may be accurately characterized in the majority of cases by subjective assessment using the IOTA group criteria. Two alternative strategies, the SRR calculator or ADNEX model may also be used in virtually all cases of adnexal tumors and might improve diagnostic performance by increasing sensitivity, thus decreasing the number of surgical interventions in pregnancy. Finally, we believe that the key to the accurate adnexal mass discrimination in pregnant women is the ability of sonographers to learn how to distinguish highly representative features of malignant lesions. Further prospective studies aiming at improved effectiveness of these methods in larger populations should be warranted. If successful, one or both scoring systems can be potentially used for stratifying patients’ risks in order to define either expectant or surgical management of adnexal complex masses in pregnancy. 

## Figures and Tables

**Figure 1 diagnostics-11-00414-f001:**
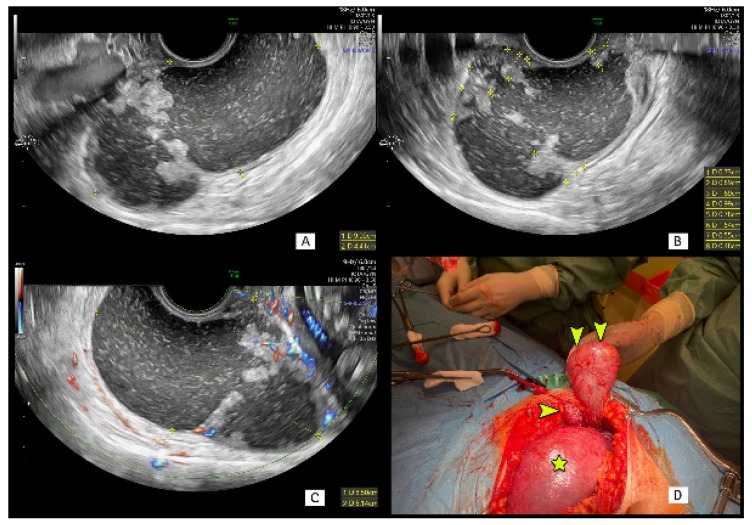
Cystic–solid lesion sonographic and surgical findings in a 36-year-old woman at 26 weeks of gestation. (**A**) Solid–cystic mass is seen in the posterior cul-de-sac with multiple internal papillary projections (**A**,**B**). Multiple blood vessels are visible in solid parts and papillary projections on color Doppler imaging (**C**). Open laparotomy at 26 weeks of gestation: small metastatic tumor (horizontal arrowhead) near gravid uterus (star, **D**) and an intact tumor capsule (vertical arrowheads, **D**). Final histology: FIGO stage IIC serous ovarian cancer.

**Figure 2 diagnostics-11-00414-f002:**
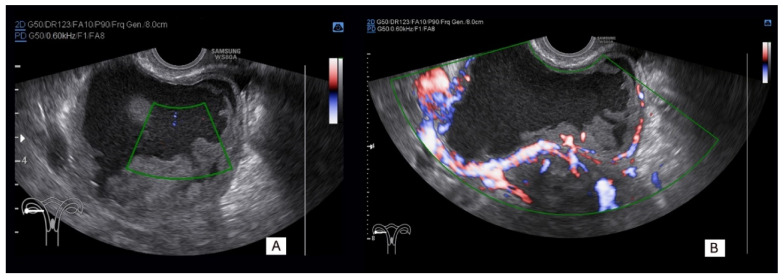
Grayscale and color Doppler findings in a 37-year-old woman with an adnexal solid–cystic unilocular tumor detected at ultrasound examination in the 17th week of gestation. Multiple internal solid parts and irregularly shaped multiple papillary projections are visible. No vascularity (**A**, green borders window) and strong vascularization (**B**, green borders window) were seen in different tumor solid parts on color Doppler imaging. Laparoscopy with a glove endobag was performed at 22 weeks of gestation, and decidualized endometrioma was confirmed on final histology.

**Figure 3 diagnostics-11-00414-f003:**
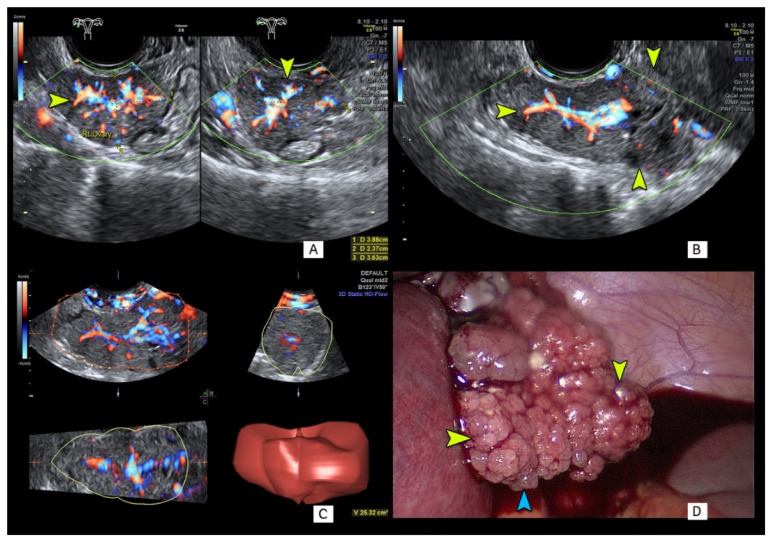
Grayscale and color Doppler findings in a 28-year-old woman with an adnexal solid tumor of 46 mm in maximum diameter detected at ultrasound examination in the eighth week of gestation. Strong vascularization of this solid mass is demonstrated with 2D color Doppler sonography (**A**,**B** show different sections of the same mass) and 3D high definition color flow imaging (**C**). Note that tumor external surface papillary projections (**D**) were not visible at ultrasound examination (blue and yellow arrows). Final diagnosis was invasive low grade serous ovarian cancer.

**Figure 4 diagnostics-11-00414-f004:**
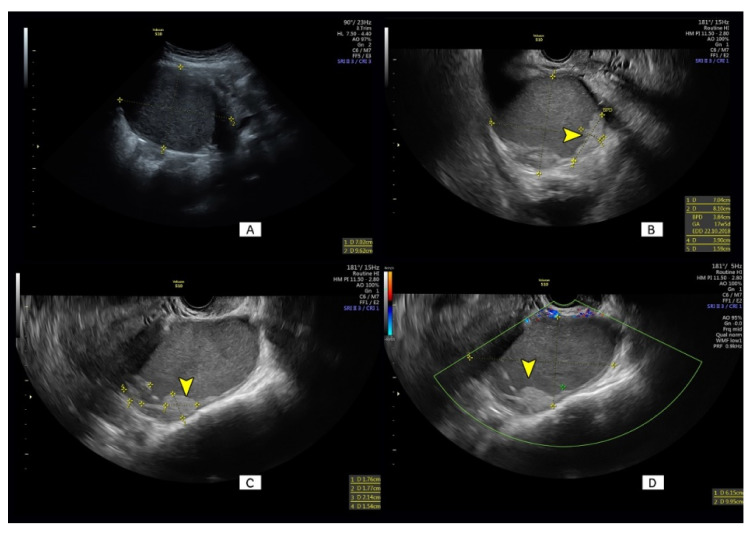
Sonographic images of an ovarian cystic–solid lesion of 96mm maximum diameter detected at 12 weeks of gestation and thought to be ovarian decidualized endometrioma (**A**). Smooth-shaped internal cyst wall papillary projections (arrowheads, **B** and **C**), with no vascularity seen at color Doppler imaging (green border window, arrowhead, **D**). Final histological diagnosis was invasive endometrioid ovarian cancer.

**Table 1 diagnostics-11-00414-t001:** Histological diagnosis of adnexal tumors in the studied group of pregnant women.

Tumor Histology	(*n* = 36)
**Benign tumors**	27
Dermoid cyst	11
Decidualized endometrioma	10
Endometrioma	1
Fibrothecoma	2
Serous cystadenoma	2
Fibroma	1
**Borderline**	
Serous	2
Invasive ovarian tumors	
Serous ovarian cancer	3
Endometrioid ovarian cancer	1
**Other**	
Colorectal cancer (metastatic)	2
Dysgerminoma	1

**Table 2 diagnostics-11-00414-t002:** Clinical characteristics of the studied group.

Characteristic	All *n* = 36 (100%)	Benign *n* = 27 (75%)	Malignant *n* = 9 (25%)	*p* Value
**Age at diagnosis (years)**	28.5 (20–42)	28 (20–42)	29 (24–41)	0.57
**Nulliparous (%)**	22 (61.1)	17 (63)	5 (55.6)	0.69
**Gestational age at diagnosis (weeks)**	13.5 (8–31)	14 (8–31)	12 (9–28)	0.8
**Gestational age at surgery (weeks)**	21.5 (12–40)	22 (14–40)	21 (12–37)	0.42
**Gestational age at delivery (weeks)**	39 (25–41)	39.5 (38–41)	38.5 (25–40)	0.06
**Mode of delivery**				0.67
Vaginal (*n*,%)	12 (33.3)	8 (29.6)	4 (44.4)	
Cesarean section (*n*,%)	20 (55.6)	16 (59.3)	4 (44.4)	
Ongoing Pregnancy (*n*, %)	4 (11.1)	3 (75)	1 (25)	

**Table 3 diagnostics-11-00414-t003:** Selected ultrasound features of the studied group of adnexal masses.

Sonographic Characteristics of Studied Tumors	All *n* = 36 (100%)	Benign *n* = 27 (75%)	Malignant *n* = 9 (25%)	*p*-Value *
**Bilateral masses**	8 (22.2)	6 (22.2)	2 (22.2)	0.64
**Maximum diameter of the lesion (mm)**	71.5 (39–206)	71 (39–206)	90 (45–135)	0.14
**Type of tumor**				0.01
Unilocular ^α^	8 (22.2)	8 (29.6)	0 (0)	
Multilocular	2 (5.5)	2 (7.4)	0 (0)	
Unilocular–solid	15 (41.7)	13 (48.2)	2 (22.2)	
Multilocular–solid	6 (16.7)	2 (7.4)	4 (44.5)	
Solid	5 (13.9)	2 (7.4)	3 (33.3)	
**Echogenicity of cyst fluid**				0.06
Anechoic	12 (33.3)	7 (25.9)	5 (55.6)	
Low level	1 (2.8)	1 (3.7)	0 (0)	
Ground glass	6 (16.7)	6 (22.2)	0 (0)	
Mixed	12 (33.3)	11 (40.8)	1 (11.1)	
Not relevant (solid mass)	5 (13.9)	2 (7.4)	3 (33.3)	
**Largest solid component (mm)**	21.5 (2–89)	18 (2–77)	39 (17–89)	0.01
**Papillary projections**	16 (44.4)	11 (40.7)	5 (55.6)	
**Number of papillary projections**				0.64
0	20 (55.6)	16 (59.3)	4 (44.5)	
1	4 (11.1)	3 (11.1)	1 (11.1)	
2	2 (5.5)	2 (7.4)	0 (0)	
3	3 (8.3)	2 (7.4)	1 (11.1)	
4	1 (2.8)	1 (3.7)	0 (0)	
>4	6 (16.7)	3 (11.1)	3 (33.3)	
**Height of the largest papillary projection (mm)**	14.5 (3–42)	12 (3–42)	15 (11–17)	0.65
**Papillation contour**				0.22
Smooth	13 (36.1)	10 (37)	3 (33.3)	
Irregegular	3 (8.3)	1 (3.7)	2 (22.2)	
**Papillation flow if papillary projections were present**				0.53
NO	25 (69.4)	20 (74.1)	5 (55.6)	
YES	11 (30.6)	7 (25.9)	4 (44.4)	
**Shadows behind papillae**				0.73
NO	33 (91.7)	24 (88.9)	9 (100)	
YES	3 (8.3)	3 (11.1)	0 (0)	
**Microcystic pattern of solid parts**				0.43
NO	34 (94.4)	26 (96.3)	8 (88.9)	
YES	2 (5.6)	1 (3.7)	1 (11.1)	
**Color Score**				0.07
1	13 (36.1)	13 (48.2)	0 (0)	
2	5 (13.9)	3 (11.1)	2 (22.2)	
3	14 (38.9)	9 (33.3)	5 (55.6)	
4	4 (11.1)	2 (7.4)	2 (22.2)	
**Ascites**	1 (2.8)	0 (0)	1 (11.1)	0.56
**Free fluid in the pouch of Douglas**				0.09
<5 mm	34 (94.4)	27 (100)	7 (77.8)	
>5 mm	2 (5.6)	0 (0)	2 (22.2)	
**Metastatatic**	2 (5.6)	0 (0)	2 (22.2)	0.09
**Ovarian crescent sign**	1 (2.8)	1 (3.7)	0 (0)	0.56

* Mann–Whitney *U* test for continuous variables and Chi-squared test for categorical variables. ^α^ denotes unilocular cyst with complex morphology found in all eight dermoid cysts operated in pregnant women. According to the IOTA group terminology, dermoid cysts are “unilocular cysts with mixed echogenicity”.

**Table 4 diagnostics-11-00414-t004:** Tumor markers concentrations and the prognostic tests results in benign and malignant masses.

Diagnostic Test	All *n* = 36 (100%)	Benign *n* = 27 (75%)	Malignant *n* = 9 (25%)	*p* Value
**CA 125 (U/mL) at diagnosis ^a^**	27 (8.6–305)	25.2 (8.6–126)	63.1 (13.5–305)	0.05
**HE4 at diagnosis ^b^**	42.2 (28.9–75.9)	41 (28.9–50.5)	50.9 (39.6–75.9)	0.03
ROMA risk (%, range) ^b^	5.3 (2.2–18.7)	5 (2.2–7.6)	8.2 (4.5–18.7)	0.04
**ROMA risk**				0.19
Low (<11.4%)	22 (91.7)	16 (100)	6 (75)	
High	2 (8.3)	0 (0)	2 (25)	
**Simple Rules Risk (SRR)**				0.03
Low (<3%)	10 (27.8)	10 (37)	0 (0)	
Intermediate (3–20%)	7 (19.4)	6 (22.2)	1 (11.1)	
High (>20%)	19 (52.8)	11 (40.8)	8 (88.9)	
**ADNEX**	11.2 (0.4–91.3)	7.6 (0.4–86.6)	52 (4.4–91.3)	0.02
**ADNEX Risk**				0.007
Low (<3%)	9 (25)	9 (33.3)	0 (0)	
Intermediate (3–20%)	12 (33.3)	10 (37)	2 (22.2)	
High (>30%)	15 (41.7)	8 (29.7)	7 (77.8)	
**ADNEX highest risk**				0.002
BENIGN *n*, %	2 (5.6)	2 (7.4)	0 (0)	
Borderline tumors (BOT) *n*, %	23 (63.9)	20 (74.1)	3 (33.3)	
FIGO stage I *n*, %	7 (19.4)	5 (18.5)	2 (22.2)	
FIGO stage II–IV, *n*,%	4 (11.1)	0 (0)	4 (44.5)	
**Subjective Assessment (SA)**				0.0001
BENIGN	20 (55.6)	19 (70.4)	1 (11.1)	
BORDERLINE	7 (19.4)	6 (22.2)	1 (11.1)	
MALIGNANT	9 (25)	2 (7.4)	7 (77.8)	

Results are presented as medians (range) or numbers and percentages. SRR—Simple Rules Risk, SA—Subjective Assessment, **^a^** CA125 not available in 9 (25%) cases, **^b^** HE4 (Human Epidydimis protein 4) and ROMA (Risk of Ovarian Malignancy Algorithm) both not available in 12 (33.3) cases.

**Table 5 diagnostics-11-00414-t005:** Predictive values of the diagnostic tests used in the study.

Test	TP	FP	FN	TN	SENS	SPEC	LR+	LR−	PPV	NPV	ACC
**CA125 ^a^**	6	5	3	13	0.67	0.72	2.40	0.46	0.55	0.81	0.70
**HE4 ^b^**	2	0	6	16	0.25	1.00	-	0.75	1.00	0.73	0.75
**ROM ^b^**	2	0	6	16	0.25	1.00	-	0.75	1.00	0.73	0.75
**ADNEX**	7	8	2	19	0.78	0.70	2.63	0.32	0.47	0.90	0.72
**SA**	8	8	1	19	0.89	0.70	3.00	0.16	0.50	0.95	0.75
**SRR**	9	17	0	10	1.00	0.37	1.59	0.00	0.35	1.00	0.53

SA—subjective assessment, SRR—Simple Rules Risk, TP—true positive, FP—false positive, FN—false negative, TN—true negative, SENS—sensitivity, SPEC—specificity, LR+—positive likelihood ratio, LR−—negative likelihood ratio, PPV—positive predictive value, NPV—negative predictive value, ACC—accuracy, **^a^** CA125 not available in nine (25%) cases, **^b^** HE4 and ROMA both not available in 12 (33.3%) cases.

## Data Availability

Data set is available in Appendix A.

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
