# Peer review of "Sonographic Assessment of Complex Ultrasound Morphology Adnexal Tumors in Pregnant Women with the Use of IOTA Simple Rules Risk and ADNEX Scoring Systems"

_diagnostics, 2021, doi:10.3390/diagnostics11030414_

Round 1

Reviewer 1 Report

Title

In line with the manuscript

Abstract

Define all the abbreviations in the abstract: IOTA, ADNEX, ROMA ...

The abstract is too long. Please check the word count and remove some sentences.

There are too many hyphenated words making it difficult to read. Please leave the words whole.

Study design: it is not known how long the study takes place, nor in how many hospitals

Manuscript

Some sentences are in italics, others are not, which makes reading unpleasant.

Sometimes there are extra spaces (line 54). Please review the entire manuscript.

The introduction is too long. Even if the rational is pretty well written, it could be shorter.

Materials and methods:

in which city and which country? (external validity)

The definition of the term IOTA appears only late in the text (line 188)

On the other hand, certain abbreviations are defined several times (SSR lines 116, 153, 212…).

Please put the internet links in references rather than in the text.

36 patients in 4 centers, for 7 years, this therefore represents approximately 1 patient per year per center…

We must therefore remain cautious about these conclusions.

Tables 2 and 3:

- the term mode of delivery is awkward when there is an item ongoing pregnancy

- The numbers in brackets represent% and sometimes min-max or quartiles? to be specified and standardized

The iconography is very interesting. On the other hand, why attach it to a patient and describe the rest of her pregnancy? it looks more like case reports than a study.

Table 4 I am annoyed by this table because ultimately not very informative concerning HE4, CA125 and ROMA. Why use averages? Besides, these are the medians that are used, why? It would have been preferable to put the number of patients who had a value higher than the recommended threshold (which have also been well defined before). *

We do not know how many patients had pathological HE4, for example (which table 5 does not tell us either, because patient can be FN in benign and in malign).

Tables 4 and 5, rather than putting an asterisk, specify in the table (n = 27 or n = 24)

What does ACC stand for? isn't it more AUC?

Discussion

The discussion is quite comprehensive regarding SA, SSR and ADNEX model.

On the other hand, it is quite poor concerning HE4, CA125 and ROMA, apart from this remain secondary objectives and therefore deserve a little more important place in the discussion. The AUC, for example, is not mentioned at all, except it remains a very good way of comparing diagnostic scores ...

References

Bibliographic references deserve to be homogenized (some have DOIs, others not., Some are incomplete (ref 5))

Finally, some useful references are missing, particularly in relation to CA125 and HE4:

Efficacy of HE4, CA125, Risk of Malignancy Index and Risk of Ovarian Malignancy Index to Detect Ovarian Cancer in Women with Presumed Benign Ovarian Tumours: A Prospective, Multicentre Trial. J Clin Med. 2019 Oct 25;8(11):1784. doi: 10.3390/jcm8111784.

Human Epididymis Protein 4 (HE4) Reference Limits in Polish Population of Healthy Women, Pregnant Women, and Women with Benign Ovarian Tumors.

Dis Markers. 2019 Aug 22;2019:3890906. doi: 10.1155/2019/3890906. eCollection 2019.

Human epididymis protein 4: factors of variation.

Clin Chim Acta. 2015 Jan 1;438:171-7. doi: 10.1016/j.cca.2014.08.020. Epub 2014 Aug 27.

Author Response

Reviewer #1

We are very thankful for a positive review.

ABSTRACT:

Query: ”Define all the abbreviations in the abstract: IOTA,ADNEX,ROMA…

Response: The abbreviations have been defined in the abstract

Query “The abstract is too long. Please check the word count and remove some sentences”

Response: The abstract has been substantially shortened and now contains 329 words.

Query: “Study design: it is not known how long the study takes place, nor in how many hospitals”

Response: The study was conducted in 3 different hospitals between 2013 and 2020.The relevant information has been added to the main manuscript text.

 Query “Manuscript. Some sentences are in italics, others are not, which makes reading unpleasant. “

Response. We apologize for that, the italics have been produced by the MDPI online system. All “italics” parts have been changed to regular text.

Query: “Sometimes there are extra spaces (line 54). Please review the entire manuscript.”

Reponse: The text has been reformatted and extra spaces have been removed in the entire manuscript

Query: “The introduction is too long. Even if the rational is pretty well written, it could be shorter.”

Response: We have shortened the Introduction and moved some text to the Discussion section and “Supplementary Materials”

Query: “Materials and methods: in which city and which country? (external validity)”

Response: The study was performed in two cities. Both cities (Lublin and Rzeszow) have been added, along with the country name-Poland

Query: “The definition of the term IOTA appears only late in the text (line 188)”

Response: The definition of the IOTA group was added in Abstract and Introduction sections

Query: “On the other hand, certain abbreviations are defined several times (SSR lines 116, 153, 212…)”.

Response: the definition of SRR was removed in the cited lines of text and now is present in the Abstract and Introduction only

Query: “Please put the internet links in references rather than in the text.”

Response: All internet links have been moved to the References section

Query: “36 patients in 4 centers, for 7 years, this therefore represents approximately 1 patient per year per center… We must therefore remain cautious about these conclusions. “

Response: Although we agree that the 36 tumors in our study is relatively small number and it could be more correct to wait for higher numbers of cases for a more significant statistical analysis, the goal was to show that adnexal tumors with complex morphology are not so rare and due to widespread use of perinatal sonographic imaging are now more frequently found in pregnant women. However, since the differentiation of such tumors with tumor markers expression or ROMA algorithm may be misleading and the correlation of tumor morphology assessed by SRR and/or ADNEX   have not been reported in pregnant women with adnexal lesions we decided to include this relatively small number of difficult cases for comparative analysis despite known limitations as suggested by the Reviewer. We agree that the conclusions must be drawn cautiously and this has been expressed at the end of the Discussion section.

Query: “Tables 2 and 3: - the term mode of delivery is awkward when there is an item ongoing pregnancy”

Response: We agree that the term “ongoing pregnancy” may look odd in that table but since the pregnancy outcome was not a primary goal of our study we also felt that it was worthwhile adding those interesting cases to the final analysis.

Query: “- The numbers in brackets represent% and sometimes min-max or quartiles? to be specified and standardized”

Response: Results are presented as medians (range) or numbers and percentages. All missing information was added to the table’s descriptions.

Query: “The iconography is very interesting. On the other hand, why attach it to a patient and describe the rest of her pregnancy? it looks more like case reports than a study.”

Response: Thank you for the positive comment on the quality of our figures. We felt that more information would highlight the difficulties with preoperative discrimination of small adnexal lesions. We have chosen to describe briefly imaging difficulties and biochemical tests results for 4 difficult and in each own way characteristic cases. However, we agree with the Reviewer that this could look more like case reports. Because of this, the patient’s history and comments were removed from all figures descriptions and moved to the Results or Discussion sections

Query: “Table 4. I am annoyed by this table because ultimately not very informative concerning HE4, CA125 and ROMA. Why use averages? Besides, these are the medians that are used, why? It would have been preferable to put the number of patients who had a value higher than the recommended threshold (which have also been well defined before). *”

Response: The medians were used because the data had skewed distribution and in such case the median is a better measure of central tendency than the mean. There is a description under Table 4: “Results are presented as medians (range) or numbers and percentages.”

Query: “We do not know how many patients had pathological HE4, for example (which table 5 does not tell us either, because patient can be FN in benign and in malign). “

For the exact characterization of the parameters tested a detailed analysis of both measurable and categorical variables as well as quantitative analysis of the medians and minimal-maximal values was performed with the categorization of variables that were both below and above the threshold values (n).  Results in the table are presented as medians (range) or numbers and percentages. False negatives cases in our study were women with malignant tumors indicated by the test as having benign lesions. The numbers are clearly indicated (Table 5, 4th column) for each test performed.

Query: “Tables 4 and 5, rather than putting an asterisk, specify in the table (n = 27 or n = 24)”

Response: Asterisks denote missing data, this is explained under the table: “*CA125-not available in 9 (25%) cases, *HE4, ROMA-both not available in 12 (33.3) cases”. Results in the table are presented as medians (range) or numbers and percentages.

Query: ”What does ACC stand for? isn't it more AUC?”

Response: “ACC” stands for the test “Accuracy” and not “AUC” which is the shortcut for the “Area Under the Curve”. We added additional lines of description “PPV-Positive Predictive Value, NPV-Negative Predictive Value, ACC-Accuracy” under Table 5.

 Query: Discussion.The discussion is quite comprehensive regarding SA, SSR and ADNEX model.On the other hand, it is quite poor concerning HE4, CA125 and ROMA, apart from this remain secondary objectives and therefore deserve a little more important place in the discussion.”

Response:  Although some missing values, the HE4, ROMA and CA125 data were included because they highlight an important clinical problem related to their limited usefulness in pregnant patients. Both existing data and the knowledge of specialists on this topic remain very poor. That is why we decided to compare these commonly used clinical tests with adnexal tumor imaging methods and more advanced, currently existing predictive systems.

We added a paragraph in the discussion section with the comments on HE4 and ROMA using the data presented in the Reviewer’s suggested references.

Query: “The AUC, for example, is not mentioned at all, except it remains a very good way of comparing diagnostic scores ... “

Response: We agree that AUC is a better way to compare the prognostic value of different diagnostic tests, however, given the limited set of data in our study we decided to use rather sensitivity, specificity, test accuracy and likelihood ratios. The latter are clinically useful measures of the test diagnostic performance as well. They use the sensitivity and specificity of the test to determine whether a test result usefully changes the probability that a condition such as ovarian cancer exists. Likelihood ratios have three main advantages: they are intuitive, they simplify the predictive value calculation and the overall evaluation of sequential testing. We are aware of disadvantages such as the non-linearity and the necessity to recalculate probabilities in odds. Although they summarize the information contained in sensitivity and specificity, these characteristics are still necessary for certain clinical decisions like choosing to perform surgery in pregnant women with ovarian tumors.

Query: “References.Bibliographic references deserve to be homogenized (some have DOIs, others not, Some are incomplete (ref 5)”

Response: We have corrected the references as suggested by the Reviewer. However some like the one from Fruhauf et al. do not have DOI number.

Query: “Finally, some useful references are missing, particularly in relation to CA125 and HE4:

Efficacy of HE4, CA125, Risk of Malignancy Index and Risk of Ovarian Malignancy Index to Detect Ovarian Cancer in Women with Presumed Benign Ovarian Tumours: A Prospective, Multicentre Trial. J Clin Med. 2019 Oct 25;8(11):1784. doi: 10.3390/jcm8111784.

Human Epididymis Protein 4 (HE4) Reference Limits in Polish Population of Healthy Women, Pregnant Women, and Women with Benign Ovarian Tumors. Dis Markers. 2019 Aug 22;2019:3890906. doi: 10.1155/2019/3890906. eCollection 2019.”

Human epididymis protein 4: factors of variation. Clin Chim Acta. 2015 Jan 1;438:171-7. doi: 10.1016/j.cca.2014.08.020. Epub 2014 Aug 27.

Response: We have added all three references and cited them promptly in the manuscript test.

Reviewer 2 Report

This is a retrospective study on characterizing masses in pregnancy. The topic is important as there is little evidence published. The authors aimed to assess the subjective assessment and the different models in characterizing ovarian masses in pregnancy. The study followed the validated classification of level of experience. The main comments are:

1- The sample is small and there is no pre study sample size assessment with power calculation. It is not known whether 36 patients are enough to generalize the findings 

2- The term"Complex masses" is not an IOTA definition to classify ovarian masses. The 5 main groups are unilocular, unilocular solid, multilocular, multilocular solid, and solid. 

3- In the inclusion criteria, complex looking masses were included but in results 8 masses with unilocular features described. There is a discrepancy here as Unilocular are considered simple rather than complex. The authors needs to redefine inclusion criteria or justify why the Unilocuar masses were included. 

4- The authors need to show how all parameters to calculate the models were used, as there were no results on missing parameters. 

Author Response

We are very thankful for a positive and detailed review that will cerainly help to improve our manuscript. In response to the Reviewer’s 2 queries here are our explanations:

Query 1- “The sample is small and there is no pre study sample size assessment with power calculation. It is not known whether 36 patients are enough to generalize the findings”

Response: Although we agree that 36 tumors in our study comprise relatively small number of cases and the findings of our study should not be generalized, the goal was to show that adnexal tumors with complex morphology are not so rare and due to widespread use of perinatal sonographic imaging are now more frequently found in pregnant women. The sample size is small but clinically it is a significant group representing epidemiologically rather rare disease such as complex ovarian masses in pregnant women.   Even though it could be more correct to wait for higher numbers of malignant cases for a more significant statistical analysis, our cases show that the differentiation of such lesions with tumor markers expression or ROMA algorithm may be misleading. Moreover,  the correlation of tumor morphology assessed by SRR and/or ADNEX   have not been reported in pregnant women with complex adnexal lesions. Given all these considerations we decided to include our relatively small number of difficult cases for comparative analysis despite known limitations as suggested by the Reviewer. As most ovarian lesions are probably examined by sonographers or specialists who are not experts in gynecologic ultrasonography, we thought that it was  reasonable to present some difficult cases of small adnexal masses and suggest that our findings could potentially offer additional clues and ideas on the performance of the different adnexal tumor risk models in daily clinical practice. Our results indicate that both SRR and ADNEX may be used with caution in the management of patients with adnexal masses detected during pregnancy, whereas CA125, HE4 and ROMA are of a very limited value.

Query 2: “The term "Complex masses" is not an IOTA definition to classify ovarian masses. The 5 main groups are unilocular, unilocular solid, multilocular, multilocular solid, and solid.”

Response: We agree that the term “complex adnexal masses” is not the IOTA group term used to describe 5 types of ovarian masses echogenicity. However, we used this term for the specific reason. Most of commonly found masses in pregnant women are simple cysts and corpus luteum cysts. Both types of lesions are not neoplastic, may disappear by the end of the 2nd trimester and are typically very easy to discriminate. Because of this, they were excluded from our study and we had to use another term for the rest of adnexal lesions that we decided to include in our analysis. In other words, we used “complex” as opposite to “simple” ultrasound morphology cysts

Query 3: “In the inclusion criteria, complex looking masses were included but in results 8 masses with unilocular features described. There is a discrepancy here as Unilocular are considered simple rather than complex. The authors need to redefine inclusion criteria or justify why the Unilocular masses were included.” 

Response: We ar grateful for this particular remark as the term was not explained. We used the term “unilocular cyst” to all our 8 cases of dermoid cysts found and operated in pregnant women. According to the IOTA group terminology most dermoid cysts are “unilocular cysts with mixed echogenicity”. Therefore, a dermoid cyst is still a unilocular one. A relevant description has been added under the Table 2. It runs as follows: ”αdenotes unilocular cyst with complex morphology (all dermoid cysts)”

Query 4: “The authors need to show how all parameters to calculate the models were used, as there were no results on missing parameters.” 

The description of the predictive models (SRR, ADNEX and ROMA) was added to the Supplementary material 2 file. The Supplementary material  section also contains our raw data stored in an MS Excel file. We are aware that no single sonographic feature or a test used in our study had more than a moderate ability to predict high risk of ovarian cancer, and thus cannot on its own be used for discriminative purposes. This does not exclude that some of the studied parameters could be of value in a multivariable analysis, or could be further used to improve diagnostic confidence.

Round 2

Reviewer 1 Report

The manuscript has been greatly improved as a result of the many comments. This is more like a trial than a case review. However, the number of patients included is quite small.

Table 4: Write in full in the table: SR Risk and SA.

Tables 4 and 5, replace "*" by "a" and "b", for example.

Reference 5 is not mentioned in the text. Is it on line 397?

Author Response

Response to reviewer’s 2 queries:

Query: “The manuscript has been greatly improved as a result of the many comments. This is more like a trial than a case review. However, the number of patients included is quite small.”

Response: We are grateful for the valuable Reviewer’s 2 comments that allowed us to significantly improve our manuscript. We agree that the number of cases is rather small, but given the importance of the topic and a willingness to improve the preoperative diagnosis of complex adnexal masses found in pregnant women we decided to present an analysis of our multicenter data. Those masses, if not operated, persist beyond second trimester of gestation, may not be easy to control by imaging methods and, when malignant, may cause significant problems during delivery.

Query: “Table 4: Write in full in the table: SR Risk and SA.”

Response: “SR Risk and SA” written in full (Simple Rules Risk and Subjective Assessment) have been placed in Table 4

Query “Tables 4 and 5, replace "*" by "a" and "b", for example.”

Response: “*” marker has been replaced by “a” and “b” according to the Reviewer’s suggestion

Query: “Reference 5 is not mentioned in the text. Is it on line 397”

Response: Reference 5 has been placed in the revised manuscript text following the statement between lines 56-58 and now runs as follows: However, the prevalence of malignancy among ovarian masses diagnosed in pregnant women varies from 0 to 9 % [5].

The statementAccording to the recently published study, the prevalence of adnexal masses in pregnancy varies between 0.15 and 5,7% [4].” as possibly redundant was removed from lines 396-397.
